# The Functional Role of Gate Loop Residues in Arrestin Binding to GPCRs

**DOI:** 10.3390/ijms262412154

**Published:** 2025-12-18

**Authors:** Sergey A. Vishnivetskiy, Daria Ghazi, Eugenia V. Gurevich, Vsevolod V. Gurevich

**Affiliations:** 1Department of Pharmacology, Vanderbilt University, Nashville, TN 37232, USA; 2Department of Psychological Sciences and Neuroscience, Belmont University, Nashville, TN 37212, USA

**Keywords:** GPCR, desensitization, arrestin, receptor binding, conformational change, structure-function

## Abstract

In all arrestins, the gate loop is the central part of the lariat loop, which has an unusual shape and participates in maintaining the basal conformation. The gate loop supplies two out of five charges that constitute a stabilizing intramolecular interaction, aspartates in the polar core between the two domains. To elucidate the functional role of individual gate loop residues, we performed comprehensive site-directed mutagenesis and tested the effects of mutations on arrestin-1 binding to its preferred target, phosphorylated light-activated rhodopsin, and unphosphorylated activated form. Out of 34 mutations tested, 24 and 25 affected the binding to phosphorylated and unphosphorylated rhodopsin, respectively. Manipulation of residues following polar core aspartates reduced preference for phosphorylated over unphosphorylated light-activated rhodopsin as dramatically as replacing these negatively charged aspartates with positively charged arginine. The data show that numerous lariat loop residues play distinct roles in arrestin-1 binding and its exquisite preference for phosphorylated light-activated rhodopsin.

## 1. Introduction

Mammals express hundreds of different G protein-coupled receptors (GPCRs), with ~800 distinct subtypes in humans [1], but only four arrestin proteins [2]. Two of these (arrestin-1 and -4) (here and below, we use systematic names of arrestin proteins. The number after the dash indicates the order of cloning: arrestin-1 (historic names: S-antigen, 48 kDa protein, visual or rod arrestin; SAG in HUGO database), arrestin-2 (β-arrestin or β-arrestin1; ARRB1 in HUGO database), arrestin-3 (β-arrestin2 or hTHY-ARRX; ARRB2 in HUGO database), and arrestin-4 (cone or X-arrestin; ARR3 in HUGO database)) are expressed in the photoreceptors in the retina, while the two non-visual subtypes (arrestin-2 and -3, also known as β-arrestin1 and 2, respectively) are ubiquitous. Activated GPCRs signal by catalyzing GDP/GTP exchange on cognate heterotrimeric G proteins [3]. The signaling of most GPCRs is stopped by a conserved two-step mechanism of homologous desensitization: active receptors are phosphorylated by specific kinases, whereupon arrestins bind active phosphoreceptors and block (arrest) their coupling to G proteins [4]. Arrestin-1 was the first member of the family discovered [5] and cloned [6]. It is expressed at very high levels in both rod [7,8] and cone [9] photoreceptors. Arrestin-1 demonstrates much greater preference for the activated phosphorylated rhodopsin (P-Rh*) over activated unphosphorylated rhodopsin (Rh*) than other arrestin family members do for the activated phosphorylated forms of their cognate receptors [10]. Vertebrate arrestins are held in their basal conformation by two intramolecular interactions, namely the polar core between the two domains, consisting of five interacting charged residues, and the three-element interaction of the C-terminus with β-strand I and α-helix I in the N-domain that involves bulky hydrophobic side chains in all three elements [11,12,13,14,15] (Figure 1). Arrestin transition into receptor-bound conformation, which was revealed by the structure of rhodopsin-bound arrestin-1 [16,17] and several structures of arrestin-2 and arrestin-3 bound to non-visual GPCRs [18,19,20,21,22,23,24,25], of both arrestin-2 and -3 bound to atypical chemikine receptor ACKR3 [26], as well as of arrestin-2 bound to the phosphopeptide representing the C-terminus of vasopressin V2 receptor [27], requires the disruption of both autoinhibitory intramolecular interactions [28].

The lariat loop (the term is from [11]; it reflects its peculiar shape, which is conserved in all arrestins crystallized thus far [11,12,13,14,15,29,30,31]) is located between β-strands XVII and XVIII of the C-domain, encompassing residues 281–322 in bovine arrestin-1. This loop is a long stretch of residues without a clear secondary structure, flanked by two very short α-helices [11] (Figure 1). Its central part (usually called the gate loop), encompassing residues 291–307 in bovine arrestin-1 (Figure 1 and Figure 2), supplies two negative charges to the polar core, including Asp296 that was shown to be critical for the selectivity of arrestin-1 for its preferred binding partner, P-Rh* [32]. Despite undisputed functional importance of the gate loop, only the role of the two polar core aspartates in it, Asp296 and Asp303 in arrestin-1, was tested experimentally with a full-length cognate receptor [32]. The role of three positively charged gate loop residues in arrestin-2 (Lys 292, Lys294, and His295, homologous to bovine arrestin-1 residues Lys298, Lys300, and His301, respectively) in binding the hyper-phosphorylated C-terminal peptide of the vasopressin V2 receptor (V2Rpp) was also tested. Here we used site-directed mutagenesis to determine the functional role of all residues in the gate loop in binding to wild type (WT) light-activated phosphorylated and unphosphorylated rhodopsin in native disk membranes. The data showed that changing fourteen out of seventeen residues has profound effects on arrestin-1 binding to rhodopsin. In particular, mutations of the residues following the polar core aspartates in the linear sequence yielded as dramatic increase in Rh* binding as mutations of these aspartates.

**Figure 1 ijms-26-12154-f001:**
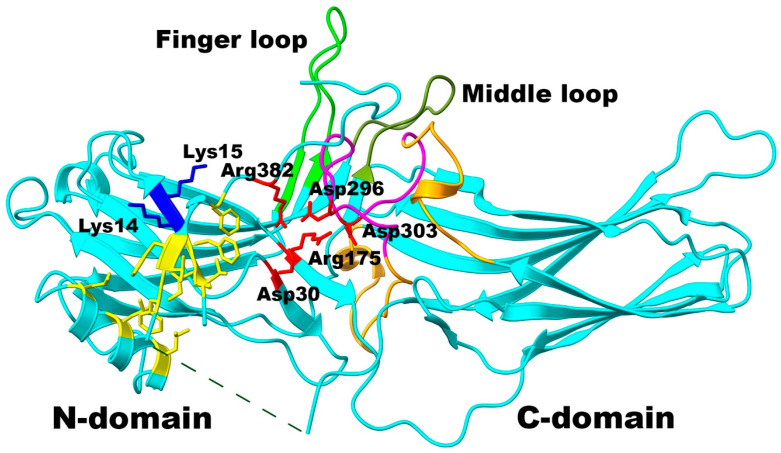
**Key elements in arrestin-1.** The structure of arrestin-1 (PDB ID 1CF1, molecule A [11]) with functionally important elements is shown as follows: the gate loop (central part of the lariat loop, residues 291–307), magenta; the remaining part of the lariat loop (residues 281–290 and 308–322), light brown; the finger loop (residues 66–81; putative activation sensor [33]), green; the middle loop (residues 132–142), olive; the lysines in β-strand I (Lys14 + Lys15, putative phosphate sensor [34,35]), dark blue stick models; polar core residues (Asp30, Arg175, Asp296, Asp303, Arg382 [11,32]), red stick models; hydrophobic residues anchoring the C-terminus to the N-domain (Val11, Ile12, Phe13 in the β-strand I; Leu103, Leu107, Leu111 in the α-helix; Phe375, Val376, Phe377, Phe380 in the C-terminus [11,36,37]), yellow stick models. Dashed line indicates long loop between β-strands XIX and XX not resolved in crystal. The image was created in UCSF ChimeraX [38,39] and labeled in Adobe Photoshop 2025 (Adobe, San Jose, CA, USA).

**Figure 2 ijms-26-12154-f002:**
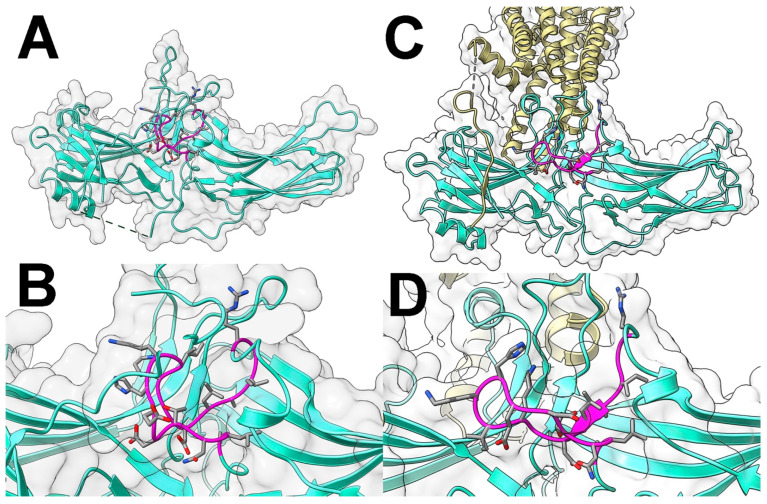
**Gate loop of arrestin-1.** (**A**) Crystal structure of basal bovine arrestin-1 (PDB ID 1CF1 [11]). (**B**) Enlarged area shows the residues targeted in this study. (**C**) Crystal structure of mouse arrestin-1–3A bound to rhodopsin (PDB ID 5W0P [17]). (**D**) Enlarged area of the complex shows the residues targeted in this study. In panels (**A**–**D**) gate loop is shown in magenta, with side chains of residues shown as stick models. The rest of the protein is shown in green. Surface of arrestin-1 is shown in pale gray. Rhodopsin in panels (**C**,**D**) is shown in yellow. Note that the numbers in mouse arrestin-1 (**D**) are N + 1 relative to the bovine protein (**B**). Images were created in UCSF ChimeraX [38,39]. The figure was assembled in Adobe Photoshop 2025 (Adobe, San Jose, CA, USA).

## 2. Results

To obtain deeper insight into the functional role of individual gate loop residues, in addition to conventional alanine scanning, we introduced charge reversals (Arg, Lys → Glu; Asp, Glu → Arg), replaced Ala with Leu that has a bulky hydrophobic side chain, and deleted glycine residues that usually break secondary structures. In addition, we replaced some residues in the bovine arrestin-1 gate loop with those found in homologous positions in other arrestin family members. Thus, the total number of tested mutations was thirty-four (Figure 3 and Figure 7).

In the direct binding assay, we used 1 nM bovine arrestin-1 produced in cell-free translation and labeled by incorporation of [^14^C] leucine. Human, mouse, and bovine arrestin-1 oligomerizes forming dimers and tetramers [40]. As experimentally determined dimerization and tetramerization constants of bovine arrestin-1 are 37 μM and 7.5 μM, respectively [41], at 1 nM concentration used in the assay, more than 99.9% of it would be monomeric. A monomer is the form competent to bind rhodopsin [41]. Thus, possible effects of introduced mutations on arrestin-1 self-association could not have affected the binding results.

We use bovine arrestin-1 residue numbers throughout for consistency: we mutated the bovine protein and there are high-resolution crystal structures of WT bovine arrestin-1 in the basal conformation (PDB 1CF1 [11] and 7JSM [29]). Note that the structure of rhodopsin-bound arrestin-1 contains mouse protein [16,17] (Figure 2), in which residue numbers are N + 1 relative to the bovine homolog.

Although Asp296Ala, Asp296Arg, Asp303Ala, and Asp303Arg were characterized previously [32], these mutants were included along with others for comparison (Figure 4). We confirmed previous reports that all four of these mutations dramatically increase arrestin-1 binding to Rh*, thereby reducing arrestin-1 preference for P-Rh* [11,32]. The effects of substitutions of Asp296 were more prominent than those of Asp303 (Figure 4 and Appendix A). Unexpectedly, alanine substitution or deletion of Gly297 yielded effects as strong as those of mutations of neighboring Asp296; Thr304Ala mutation yielded a phenotype similar to that of proteins with Asp303 substitutions (Figure 4 and Appendix A). The effects of manipulation of Gly297 suggest that changes in the gate loop conformation destabilize the polar core as effectively as the replacement of the adjacent negatively charged Asp296.

Both substitutions of Arg291 (Arg291Ala and Arg291Glu) increased the binding to P-Rh* by 20–30% and more than doubled the binding to Rh* (Figure 4), thereby reducing arrestin-1 selectivity for P-Rh* (Appendix A). The deletion of Gly292 increased P-Rh* and Rh* binding by 38% and 156%, respectively, while Gly292Ala mutation did not affect P-Rh* binding but enhanced Rh* binding by ~60% (Figure 4). These differential effects suggest that the change in gate loop conformation, which deletion affects more significantly than alanine substitution, is the key. Both substitutions reduced arrestin-1 preference for P-Rh* (Appendix A). Interestingly, dramatic changes in the size of the side chain of the next two residues (Ile293Ala and Ala294Leu) did not significantly affect arrestin-1 function (Figure 4). Lys298Ala substitution slightly increased P-Rh* binding, while Lys298Glu reduced it.

The Ile299Ala mutation reduced the binding to P-Rh*, but not to Rh* (Figure 4). The structure of the complex [17] suggests that Lys300 engages phosphorylated Ser338 in the C-terminus of rhodopsin. Charge reversal would preclude this interaction. However, the strong negative impact on P-Rh* and Rh* binding, particularly on Rh*, suggests that possible interaction with rhodopsin-attached phosphate (which is absent in Rh*) does not underlie this effect. Notably, alanine substitution of Lys300 did not reduce the binding to WT P-Rh* (Figure 4), suggesting the limited role of this interaction. Both substitutions of His301 facilitated Rh* binding, but only His302Glu enhanced the binding to P-Rh* (Figure 4). Alanine substitution of Glu302 similarly reduced the binding to both forms of rhodopsin, whereas charge reversal by Glu302Arg substitution slightly increased the binding to P-Rh*, but not to Rh* (Figure 4).

Mutations can change thermal stability of proteins, which would affect the results of the binding tests. Although the yields of soluble translated mutants did not reveal any significant changes (Appendix A), we additionally tested thermal stability by incubating the proteins at assay temperature (37 °C) for up to 30 min, which is six times longer than the binding assay (Figure 5). Previously, we found that charge reversals of the residues forming the polar core (Figure 1) affect the thermal stability of arrestin-1 most dramatically [42]. Therefore, we tested D296R and D303R mutants as well as their functional mimics G297A, ΔG297, and T304A (Figure 4). We found that at 37 °C arrestin-1-D296R was somewhat less stable than WT (*p* < 0.01, as determined by time–protein interaction by two-way ANOVA analysis), whereas the stability of other mutants was not significantly different from WT (Figure 5). Thus, the difference in thermal stability was unlikely to significantly affect the results of the binding assay (Figure 4). Out of 52 measurements of mutant binding (Figure 4), 25 showed an increase, 17 showed no change, and only 10 showed a decrease, which supports this notion.

Based on sequence alignment of arrestin proteins from various species (Figure 6), we introduced several additional mutations in the gate loop of arrestin-1 to assess possible functional effects of non-conserved residues in these positions in some members of the arrestin family. These included Arg291Gln (as in bovine and mouse arrestin-4), Ile293Leu, Ile299Leu, Ile293Leu + Ile299Leu (as in mammalian arrestin-2, -3, and -4, as well as in arrestins from *C. intestinalis*, *C. elegance*, and *Drosophila kurtz*), His301Gln, Glu302Gly, His301Gln + Glu302Gly (as in bovine arrestin-4), and Lys298Gln (as in mouse arrestin-3, and arrestins from *C. elegans*, *C. intestinalis*, and *Drosophila kurtz*). For the sake of comparing these substitutions with previously tested ones in the same experiment, we repeated the binding of relevant mutants shown in Figure 4 along with new ones, changing the residues in bovine arrestin-1 to those found in homologous positions in other family members (Figure 6).

Arrestin-4 demonstrates a significantly lower preference for the phosphorylated form of its cognate receptor over the unphosphorylated than arrestin-1 does [14]. As expected, the replacement of Arg291 native for arrestin-1 with Gln characteristic for arrestin-4 reduced arrestin-1 preference for P-Rh* over Rh*, similar to Arg291Ala substitution (Figure 7 and Appendix A). Thus, the presence of Gln in some arrestin-4 proteins likely reduces the selectivity for phosphorylated receptor as compared to Arg in arrestin-1 but does not significantly affect the ability to bind the receptor. The effects of replacing Ile293 and Ile299 found in human, bovine, and mouse arrestin-1 with Leu turned out to be similar to alanine substitutions: both Leu mutants yielded somewhat lower binding to P-Rh*, with Ile293Ala also reducing the binding to Rh* (Figure 7). The Ile299Leu mutant demonstrated the same selectivity as WT, but the selectivity of Ile293Leu was unexpectedly increased (Appendix A). However, when both Ile residues were replaced with Leu simultaneously, as in vertebrate arrestin-2, -3, -4, and invertebrate arrestins (Figure 6), rhodopsin binding of the double mutant and its selectivity for P-Rh* was not significantly different from WT arrestin-1 (Figure 7 and Appendix A), suggesting that these two Leu residues in other subtypes might play a role in the function(s) that arrestin-1 does not have. The mutant with the substitution of positively charged Lys298 for uncharged Gln (as in some mammalian and all invertebrate arrestins; Figure 6) demonstrated WT level binding to Rh* and slightly lower binding to P-Rh* (Figure 5), but no significant change in selectivity (Appendix A). Notably, the Lys298Gln mutant displayed lower binding to both forms of rhodopsin than Lys298Ala (Figure 7). Thus, it appears that the presence of uncharged Gln in position 298 instead of Lys in arrestin-1 does not dramatically affect receptor interactions. Interestingly, the effects of neutral Gln in position 298 were similar to those of negatively charged Glu (Figure 4 and Figure 7), suggesting that H-bonding, rather than charge–charge interaction involving this residue, is functionally important. This hypothesis is supported by the finding that the substitution of Lys298 for an alanine (small side chain lacking H-bonding capability) increased the binding to P-Rh* (Figure 4 and Figure 7). Replacing His301 with Gln in this position (as in invertebrate arrestins; Figure 6) enhanced the binding to both forms of rhodopsin, but more strongly to Rh*, thereby reducing arrestin-1 preference for P-Rh* (Appendix A). The effect was even stronger than in the case of His301Ala substitution (Figure 4, Figure 7 and Appendix A). Arrestin-1 with glycine in position 302 (as in bovine arrestin-4, instead of Glu conserved in almost all other family members; Figure 6) showed somewhat reduced P-Rh* binding but bound Rh* like WT (Figure 7). The negative impact of Glu302Ala substitution is stronger than that of Glu302Gly, reducing the binding to both P-Rh* and Rh* (Figure 4 and Figure 7). Arrestin-1 with double mutation His301Gln + Glu302Gly (as in bovine arrestin-4; Figure 6) bound P-Rh* like WT, but demonstrated greatly enhanced Rh* binding, similar to the single His301Gln mutant (Figure 7), with corresponding reduction in selectivity (Appendix A). Apparently, His301Gln substitution largely determined the effect observed in the double mutant. Notably, the performance of His301Gln + Glu302Gly resembled that of arrestin-4 [14], of which these mutations were designed to mimic (Figure 7).

## 3. Discussion

Vertebrate arrestins are held in their basal conformation by two intramolecular interactions, namely the polar core between the two domains, and three-element interaction of the C-terminus with the β-strand I and α-helix I in the N-domain [11,12,13,14,15]. Both must be disrupted for arrestin transition into receptor-binding conformation [28]. The central part of the lariat loop, often called the gate loop (Figure 1), in all arrestins supplies two out of five charges to the polar core, which is unusual for a soluble protein arrangement of five interacting charged side chains that are virtually solvent-excluded [11]. The peculiar shape of the lariat loop, including its central part, the gate loop, is conserved in the basal conformation of all four subtypes of vertebrate arrestins [11,12,13,14,15,29,31]. Its conformation in receptor-bound arrestin-1 [16,17], -2 [18,19,20,21,22,23,24,25,26,57,58], and -3 [25,26], as well as in the receptor bound-like conformation of arrestin-3 in the trimer [59], is significantly different. Thus, the gate loop likely plays an important role in the conformational rearrangement accompanying receptor binding. Indeed, the reversal of the polar core negative charges supplied by the lariat loop “pre-activated” arrestins, yielding mutants that readily bind active unphosphorylated forms of their cognate receptors [32,37,60,61,62]. However, this critically important element of arrestins remains understudied: the functional role of the other residues in the gate loop was never tested experimentally. Unfortunately, the only comprehensive studies of the effects of alanine substitutions of all arrestin-1 residues on P-Rh* [63] and phospho-opsin [64] binding used a relatively low-sensitivity assay and yielded inconclusive results. The role of three positively charged residues in the gate loop of arrestin-2 (Lys292, Lys294, and His295, homologous to bovine arrestin-1 residues Lys298, Lys300, and His301, respectively; Figure 6) in binding to the hyper-phosphorylated C-terminal peptide of vasopressin V2 receptor (V2Rpp) was recently tested using the rate of hydrogen–deuterium exchange (HDX) in different arrestin-2 elements [65]. HDX yields the information on the accessibility of the arrestin molecule to the solvent, from which the effects on the binding can be only deduced. Our assay yields the information on the binding, from which structural effects of mutations can be only deduced. Thus, below, we compare the deductions from the results obtained by two different methods. The authors of the HDX study [65] found that Lys294 directly participates in the binding of a phosphate on V2Rpp but does not affect the overall conformation of the V2Rpp-bound arrestin-2. The other two residues, Lys292 and His295, were found to contribute to the stability of the polar core in the basal conformation and to affect the ability of the finger loop to assume a specific conformation upon V2Rpp binding [65]. Our data suggest that the functional roles of these positive charges in the gate loop of arrestin-1 in the binding to WT rhodopsin are not always the same as in arrestin-2 binding to V2Rpp. Lys300Ala substitution (Lys300 in arrestin-1 is homologous to Lys294 in arrestin-2) did not affect P-Rh* and Rh* binding, whereas Lys300Glu mutation reduced the binding to both forms of rhodopsin (Figure 4), while increasing arrestin-1 preference for P-Rh* (Appendix A). We earlier showed that this lysine in bovine and mouse arrestin-1, as well as its homologues in bovine non-visual arrestin-2 and -3, likely plays a role in the binding to full-length cognate receptors, but not specifically in the binding of a receptor-attached phosphate [34]. Charge reversal by Lys298Glu (homolog of Lys292 in arrestin-2) reduced the binding to P-Rh* without affecting Rh* interaction (Figure 4). This is consistent with its interaction with receptor-attached phosphate absent in Rh*, but not with its role in receptor-binding-induced conformational rearrangements in arrestin proteins. Both substitutions of His301 significantly increased Rh* binding (Figure 4), thereby decreasing arrestin-1 preference for P-Rh* (Appendix A). This is consistent with the role of this residue in arrestin transition into receptor-bound conformation, as suggested in [65] for arrestin-2.

Arrestin-1 demonstrates much greater preference for P-Rh* than other family members for the active phosphorylated form of their cognate receptors (reviewed in [10]). Among 34 mutations tested here, thirteen did not significantly affect this selectivity, and seven even increased it, whereas the majority (19 mutations) decreased it (in many cases, dramatically) (Figure 4, Figure 7, Appendix A). Our recent studies suggest that the function of many native residues in arrestin-1 is to maintain high selectivity for P-Rh* by suppressing Rh* binding [66,67]. Thus, the data raise the question of why evolution did not favor selectivity-enhancing residues in arrestin proteins (see alignments in [2,68]). Among the seven substitutions that increased the preference for P-Rh* over Rh*, five reduced absolute P-Rh* binding; so in these cases, the answer appears to be clear: these are detrimental for the binding to the preferred arrestin-1 partner, P-Rh* (Figure 4). However, other selectivity enhancers, Ala294Leu and Leu306Ala, did not reduce P-Rh* binding (Figure 4). However, the analysis of multiple arrestins from a variety of vertebrate and invertebrate species [68] showed that leucine in position homologous to Ala294 is not found in arrestins. All analyzed homologs harbor an alanine in a position equivalent to Ala294, as well as Leu in a position homologous to Leu306 in bovine arrestin-1 (Figure 6). It is tempting to speculate that Ala and Leu in these positions are advantageous for the thermal stability (which we tested only for 30 min at 37°). WT arrestin-1 is remarkably stable, and it did not lose activity even after incubation for hours at 39 °C [42,69].

The effects of substitutions mimicking other arrestin subtypes yielded expected answers only in some cases. Among arrestin-4-inspired mutations (Figure 6), His301Gln dramatically changed the selectivity of arrestin-1 towards that of arrestin-4 (Appendix A), and Arg291Gln acted in the same direction to a lesser extent, whereas individual Glu302Gly substitution did not change selectivity (while reducing the binding to both forms of rhodopsin). Importantly, double mutant His301Gln + Glu302Gly mimics bovine arrestin-4; these residues in mouse and human arrestin-4 are the same as those in arrestin-1 (Figure 6) and arrestin-4 was less selective than parental arrestin-1, as expected (Figure 7 and Appendix A). Lys298Gln, as well as double substitution Ile293Leu + Ile299Leu, only marginally affected receptor binding (Figure 7 and Appendix A), suggesting that these residues might play a role in other functions of vertebrate non-visual and invertebrate arrestins. It should be noted, however, that a relatively long side chain with H-bonding capability was preserved by Lys298Gln substitution, while in the latter case, residues with bulky hydrophobic side chains were substituted with alternatives of similar size and chemical nature. Thus, receptor binding might require these chemical characteristics, rather than residue identity.

Overall, the conservation of the sequence of the gate loop in arrestins from round worm *C. elegans*, fly *Drosophila melanogaster*, tunicate *Ciona officinalis*, squid *Loligo pealei*, and mammals, i.e., animals that had common ancestors more than 600,000,000 years ago, is striking: amongst seventeen residues in the gate loop, eleven are strictly conserved, and the other six are either conserved in most, or underwent conservative substitutions (Figure 6). This sequence contains seven residues carrying charges: three with bulky hydrophobic side chains, two glycines, two alanines with very small side chains, and only three residues that do not belong to either of these categories. Available structures suggest that charged and bulky hydrophobic side chains in the core of proteins usually mediate intramolecular interactions; glycines disallow the formation of β-strands and α-helices, whereas alanines preclude steric clashes with nearby elements due to the small size of the side chain.

Both polar core aspartates and several other lariat loop residues contribute to exceptional selectivity of arrestin-1 for P-Rh* over Rh*. However, the positioning of the side chains with different chemical nature in the finger loop is virtually identical in arrestins with high selectivity (arrestin-1 from different species) and in less selective isoforms, such as non-visual or cone arrestins, and even in invertebrate visual arrestins that bind unphosphorylated rhodopsins [70,71,72]. Thus, it appears likely that the function of the gate loop goes beyond proper positioning of the polar core aspartates. Its conserved sequence and unusual shape appear to be important for the function shared by all arrestin proteins—the binding to GPCRs.

## 4. Materials and Methods

*Materials*. [γ-^32^P]ATP and [^14^C]leucine were from Perkin–Elmer (Waltham, MA, USA). Restriction endonucleases, Vent DNA polymerase, and Quick T4 DNA ligase were from New England Biolabs (Ipswich, MA, USA). Rabbit reticulocyte lysate was made in bulk by Ambion (Austin, TX, USA). SP6 RNA polymerase was overexpressed in *E. coli* and purified, as described [73]. DNA purification kits (mini-, midi-, and maxi; 3 mL, 50 mL, and 100 mL of bacterial culture, respectively) were from Zymo Research (Irvine, CA, USA). All other reagents were from Sigma-Aldrich (St. Louis, MO, USA).

*Mutagenesis and plasmid construction*. For in vitro transcription by SP6 RNA polymerase bovine arrestin-1 cDNA was subcloned into pGEM2 vector (Promega; Madison, WI, USA) with “idealized” 5-UTR that ensures efficient translation of uncapped mRNAs [73] between Eco RI and Hind III sites, as described [74]. Mutations were introduced by PCR. Unique restriction sites Msc I (codons 284–286) and Spe I (codons 323–325) engineered by the introduction of the silent mutation coding sequence of bovine arrestins-1 (described in [75]) were used to subclone generated mutant fragments. All mutations were confirmed by dideoxy sequencing (GenHunter Corporation, Nashville, TN, USA).

In vitro transcription [73], translation, and calculation of specific activity of synthesized radiolabeled arrestin proteins [76], preparation of unphosphorylated and phosphorylated bovine rhodopsin, and quantification of the level of its phosphorylation [77,78] were performed as described. Translation yields and the fraction of translated protein that remains in the supernatant after 1 h centrifugation at 357,000× *g* (to remove ribosomes and aggregated proteins) (100,000 rpm in TLA120.1 rotor) suggest that none of the mutants used had folding problems (Appendix A). Experiments in vitro [79] and in vivo [80] demonstrate that three rhodopsin-attached phosphates are necessary for tight arrestin-1 binding. P-Rh* used in binding experiments had on average 2.6 phosphates per rhodopsin, which suggests that a large fraction of it carries three or more phosphates. To ensure that rhodopsin is not rate-limiting, very high molar excess was used in the assay. Rhodopsin was phosphorylated in bright light by endogenous rhodopsin kinase (systematic name GRK1), as described [81]. Thus, it is likely that biologically relevant sites (the C-terminus of bovine rhodopsin has seven phosphorylatable serines and threonines [82,83]) were phosphorylated. Phosphorylated rhodopsin has been regenerated by two additions of three-fold molar excess of 11-cis-retinal, as described [76].

*Direct binding assay* was performed, as described [76]. Briefly, 1 nM arrestin-1 (50 fmol, specific activity 12.1–12.6 dpm/fmol) was incubated with 0.3 μg of P-Rh* or Rh* (7.8 pmol; concentration in the assay 156 pM) in 50 mL of 50 mM Tris-HCL, pH 7.4, 100 mM potassium acetate, 1 mM EDTA, and 1 mM DTT for 5 min at 37 °C under room light. Samples were then cooled on ice for 2–3 min, whereupon bound arrestin-1 was separated from free arrestin-1 and unincorporated [^14^C] leucine at 4 °C by gel filtration chromatography on 2 mL column of Sepharose CL-2B. Bound arrestin-1 eluted with rhodopsin-containing membranes was quantified by liquid scintillation counting on Tri-Carb (PerkinElmer, Waltham, MA). Non-specific “binding” (likely reflecting the aggregation of arrestin-1 proteins during the assay) was determined in samples without rhodopsin and subtracted.

*Thermal Stability Test.* Translation mixes obtained after centrifugation were kept on ice (0, control) or incubated at assay temperature (37 °C) for 5, 15, and 30 min and then cooled on ice. The binding to P-Rh* of all samples was performed, as described above.

*Data Analysis and Statistics.* Statistical significance of the differences between mutants and WT arrestin-1 in each group was determined by one-way ANOVA (analysis of variance) with Dunnett’s multiple comparison test using Prism 10 software (GraphPad, Boston, MA). Stability data were analyzed by two-way ANOVA, where protein–time interaction reflected the difference in the survival rate. *p* values < 0.05 were considered statistically significant and indicated, as follows: * *p* < 0.05; **, *p* < 0.01; *** *p* < 0.001.

## Figures and Tables

**Figure 3 ijms-26-12154-f003:**
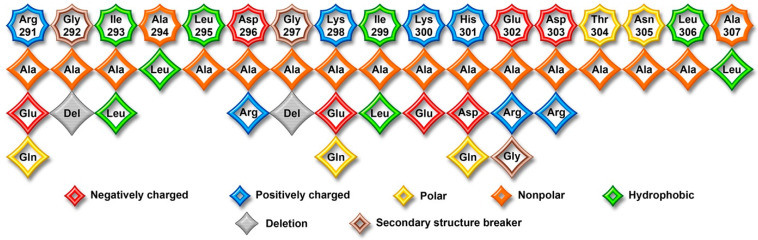
**The native sequence of the gate loop of bovine arrestin-1 and mutations introduced.** All residues are color coded, as follows: small non-polar, orange; uncharged polar, yellow; positively charged, blue; negatively charged, red; hydrophobic, green. Glycine residues breaking secondary structures are shown in brown. Deletions are shown in gray. Native sequence is shown in hexagons, replacing residues in rhombs.

**Figure 4 ijms-26-12154-f004:**
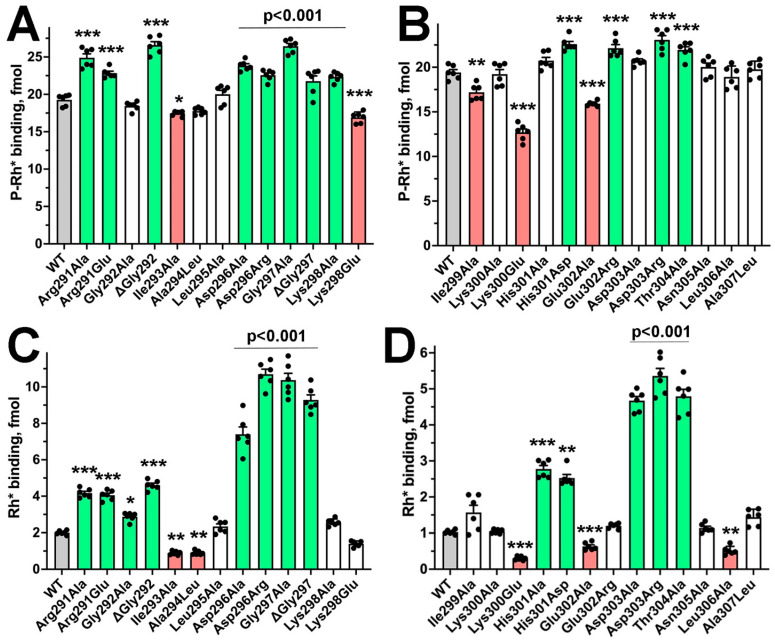
**The effect of gate loop mutations on arrestin-1 binding to rhodopsin.** The binding of indicated mutants of arrestin-1 to P-Rh* (**A**,**B**) and Rh* (**C**,**D**) was determined using radiolabeled arrestins, produced in cell-free translation, in the direct binding assay with purified phosphorylated or unphosphorylated light-activated bovine rhodopsin, as described in Methods. Small black circles represent individual measurements (*n* = 6). The binding to P-Rh* and Rh* was analyzed separately in each of the two groups. Statistical significance of the differences between WT arrestin-1 and mutants was determined by one-way ANOVA followed by Dunnet post hoc comparison to WT with correction for multiple comparisons. Statistical significance (*p* value) is either shown (directly applicable to all bars under the line) or indicated, as follows: *, *p*  <  0.05; **, *p* < 0.01; ***, *p* < 0.001 to WT. Bars corresponding to mutants with increased or decreased binding are colored light green and light red, respectively; uncolored bars show no significant difference from WT. Bars corresponding to WT are gray. Panels were created by Prism 10 (Graphpad software, Inc., San Diego, CA, USA). The figure was assembled in Adobe Photoshop 2025 (Adobe, San Jose, CA, USA).

**Figure 5 ijms-26-12154-f005:**
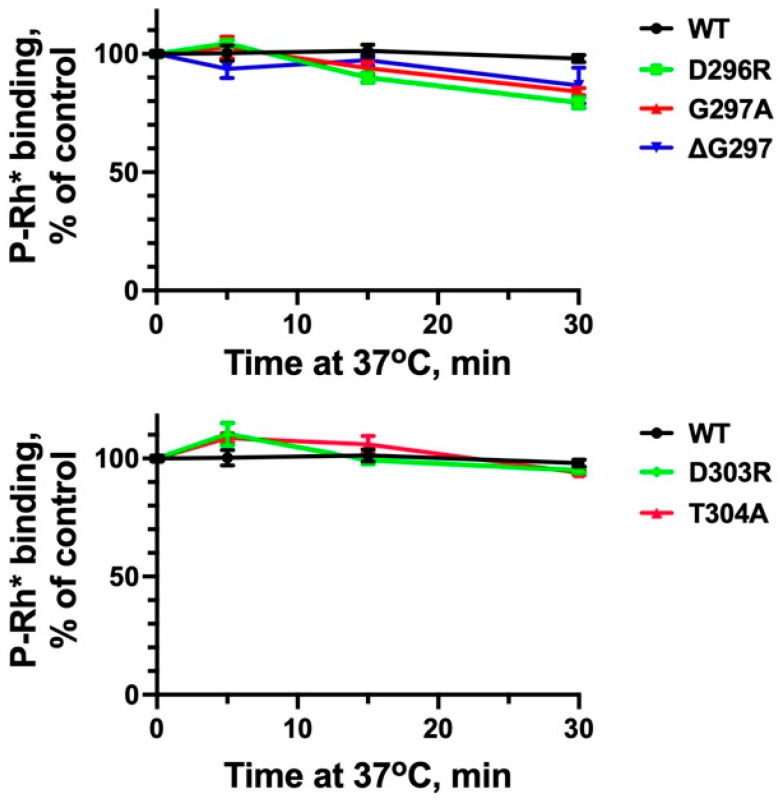
**Thermal stability of mutants.** Translation mix containing WT arrestin-1 and indicated mutants was kept on ice (0, control) or incubated at 37 °C for indicated times and then cooled on ice, whereupon the binding of all samples to P-Rh* was measured. Plotted survival curves (as % of control) show means ± SEM of three independent experiments. Statistical analysis of the binding data (in fmol bound) showed that the stability (protein–time interaction in two-way ANOVA) of only D296R mutant was different from WT (*p* < 0.01). Statistical analysis was performed and panels were created in Prism 10 (Graphpad software, Inc., San Diego, CA, USA). The figure was assembled in Adobe Photoshop 2025 (Adobe, San Jose, CA, USA).

**Figure 6 ijms-26-12154-f006:**
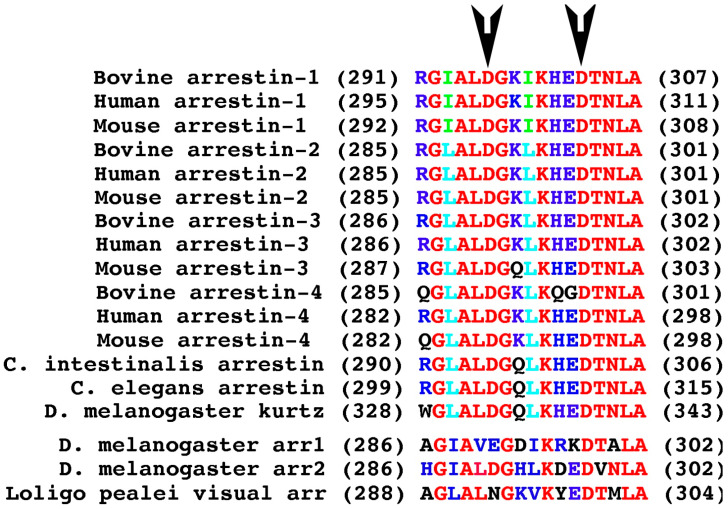
**Conservation of the gate loop sequence.** The numbers of the first and last residue in each arrestin are shown in parentheses before and after the sequence in single-letter code, respectively. Conserved residues are shown in red; conservative substitutions are shown in light blue; residues conserved in most, but not all, arrestins are shown in dark blue; residues conserved only in arrestin-1 from different mammalian species are shown in green. Three visual arrestins from two invertebrate species, fruit fly *Drosophila melanogaster* and squid *Loligo pealei*, as well as arrestins from the round worm *C. elegans* and tunicate *C. intestinalis*, are shown for comparison. Black arrows point to the two aspartates of the polar core. The sequences are from arrestin-1 bovine [6], human [43], mouse [44], arrestin-2 bovine [45], human [46], mouse [47], arrestin-3 bovine [45], human [48], mouse [47], arrestin-4 bovine [49], human [50], mouse (GeneBank AF156979), *Drosophila* arrestin1 [51], arrestin2 [52], squid visual arrestin [53], *Drosophila* non-visual arrestin *kurtz* [54], *Caenorhabditis elegans* arrestin [55], and *Ciona intestinalis* arrestin [56]. The figure was created in Adobe Photoshop 2025 (Adobe, San Jose, CA, USA).

**Figure 7 ijms-26-12154-f007:**
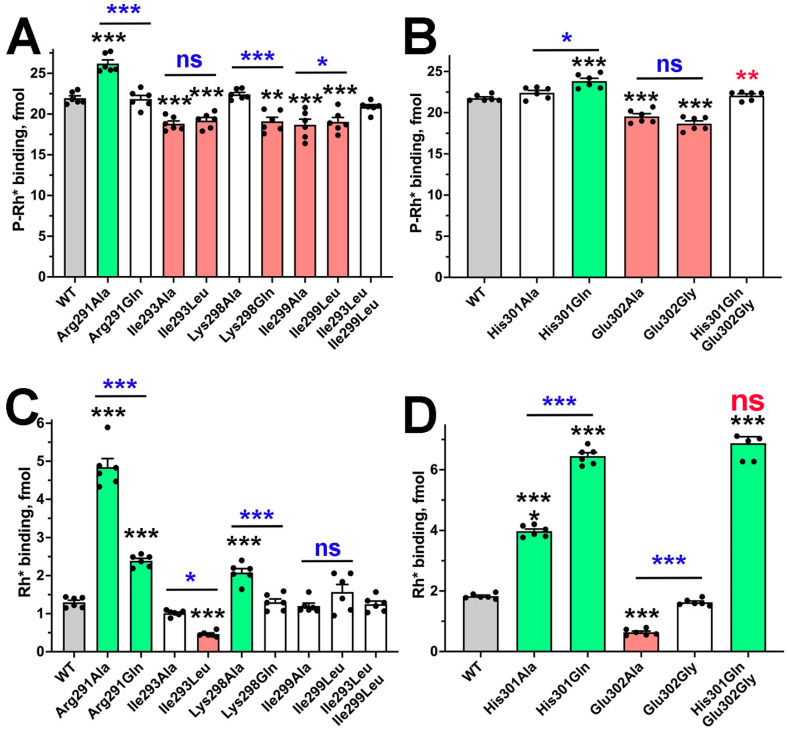
**Mutations in the gate loop of bovine arrestin-1 mimicking other subtypes.** The binding of indicated mutants of arrestin-1 to P-Rh* (**A**,**B**) and Rh* (**C**,**D**) was determined using radiolabeled arrestins, produced in cell-free translation, in the direct binding assay with purified phosphorylated or unphosphorylated light-activated bovine rhodopsin, as described in Methods. Small black circles represent individual measurements (*n* = 6). The binding to P-Rh* and Rh* was analyzed separately in each of the two groups. Statistical significance of the differences between WT arrestin-1 and mutants was determined by one-way ANOVA followed by Dunnet post hoc comparison to WT with correction for multiple comparisons. Statistical significance (*p* value) is indicated, as follows: ns, not significant; *, *p*  <  0.05; **, *p* < 0.01; ***, *p* < 0.001 to WT. The signs over the lines indicate the comparison of values under the line. The results of pairwise comparisons between indicated mutants are shown in blue, the comparisons of His301Gln + Glu302Gly with His301Gln are shown in red. Bars corresponding to mutants with increased or decreased binding are colored light green and light red, respectively; uncolored bars show no significant difference from WT. Bars corresponding to WT are gray. Panels were created by Prism 10 (Graphpad software, Inc., San Diego, CA, USA). The figure was assembled in Adobe Photoshop 2025 (Adobe, San Jose, CA, USA).

## Data Availability

The data are presented in the manuscript. Raw binding data obtained in each experiment are available upon request.

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
