# Peer review of "The Functional Role of Gate Loop Residues in Arrestin Binding to GPCRs"

_ijms, 2025, doi:10.3390/ijms262412154_

Round 1
Reviewer 1 Report
Comments and Suggestions for Authors
The manuscript entitled “The functional role of the gate loop residues in arrestin binding to GPCRs” by Vishnivetskiy and coworkers presents a comprehensive and systematic site-directed mutagenesis study to investigate the functional role of residues within the gate loop of arrestin-1. The authors evaluated 34 mutations across 17 residues, focusing on their effects on binding to both phosphorylated (P-Rh) and unphosphorylated light-activated rhodopsin (Rh). The study provides a detailed functional map of this critical structural element and offers valuable insights into the molecular basis of arrestin-1’s selectivity for phosphorylated receptors. However, I have some concerns about the manuscript. I hope addressing those question might help the authors to improve the manuscript.
The background section does not include a recent relevant study:
A 2024 study in Structure. 2024;32:1358–1366, which also examines the role of gate loop residues. This work should be cited in the Introduction and/or Discussion to properly situate the current study within the latest research landscape. A brief comparison with the findings of that study would strengthen the manuscript.
Gate loop, middle loop, and finger loop could be indicated in Figure 1 to clearly clarify the locations of those previously studied ones.
The quantitative analysis of phosphorylation selectivity is insufficient. While Figures 4 and 7 present binding data for P-Rh and Rh separately, the manuscript lacks a direct, quantitative visualization of selectivity changes. Some suggestion might help you strengthen the conclusions: Calculate and display the P-Rh/Rh binding ratio for each mutant as a dedicated figure panel or as supplementary material. Or, you might present these ratios as a scatter plot or bar chart with statistical significance markers to clearly show which mutations enhance, maintain, or reduce arrestin-1's preference for phosphorylated rhodopsin. This would provide a more intuitive and compelling demonstration of the dramatic selectivity shifts described in the text.
The thermal stability dataset is not completed. Figure 5 presented thermal stability data for only 5 mutants (D296R, G297A, ΔG297, D303R, T304A). To definitively rule out stability artifacts as a confounding factor in binding assays, data for the other mutants should be provided, even if no significant changes were observed for them.
Author Response
Comment 1. The manuscript entitled “The functional role of the gate loop residues in arrestin binding to
GPCRs” by Vishnivetskiy and coworkers presents a comprehensive and systematic sitedirected
mutagenesis study to investigate the functional role of residues within the gate
loop of arrestin-1. The authors evaluated 34 mutations across 17 residues, focusing on their
effects on binding to both phosphorylated (P-Rh) and unphosphorylated light-activated
rhodopsin (Rh). The study provides a detailed functional map of this critical structural
element and offers valuable insights into the molecular basis of arrestin-1’s selectivity for
phosphorylated receptors. However, I have some concerns about the manuscript. I hope
addressing those question might help the authors to improve the manuscript.
Response 1. We appreciate reviewer’s help and believe that addressing expressed concerns improved the manuscript.
Comment 2. The background section does not include a recent relevant study:
A 2024 study in Structure. 2024;32:1358–1366, which also examines the role of gate loop
residues. This work should be cited in the Introduction and/or Discussion to properly situate
the current study within the latest research landscape. A brief comparison with the findings
of that study would strengthen the manuscript.
Response 2. We apologize for this omission. Even though the authors studied arrestin-2 binding to hyper-phosphorylated C-terminus of V2 vasopressin receptor (V2Rpp), whereas we determined the binding of arrestin-1 mutants to WT rhodopsin, we agree that the results are relevant for arrestin-1. We mentioned this study (Kim, Ashim, Ham, Yu, & Chung, 2024) in the introduction and compared authors’ conclusions regarding arrestin-2 with our data on arrestin-1 in the discussion. Our data suggest that the roles of homologous positive charges in the gate loop of arrestin-1 in its binding to full-length rhodopsin and in arrestin-2 binding to V2Rpp are similar.
Kim, K., Ashim, J., Ham, D., Yu, W., & Chung, K. Y. (2024). Roles of the gate loop in beta-arrestin-1 conformational dynamics and phosphorylated receptor interaction. Structure, 32(9), 1358-1366.
Comment 3. Gate loop, middle loop, and finger loop could be indicated in Figure 1 to clearly clarify the locations of those previously studied ones.
Response 3. We thank the reviewer for this excellent suggestion! In the revised Figure 1 we highlighted the middle loop (residues 132-142), as requested. Note that gate and finger were already highlighted. Fo the sake of clarity we also labeled the finger and middle loop.
Comment 4. The quantitative analysis of phosphorylation selectivity is insufficient. While Figures 4 and 7 present binding data for P-Rh and Rh separately, the manuscript lacks a direct,
quantitative visualization of selectivity changes. Some suggestion might help you
strengthen the conclusions: Calculate and display the P-Rh/Rh binding ratio for each
mutant as a dedicated figure panel or as supplementary material. Or, you might present
these ratios as a scatter plot or bar chart with statistical significance markers to clearly show
which mutations enhance, maintain, or reduce arrestin-1's preference for phosphorylated
rhodopsin. This would provide a more intuitive and compelling
Response 4. We thank the reviewer for an excellent suggestion! We calculated P-Rh*/Rh* binding ratios for all mutants and presented them in supplemental Figures S1 (related to Fig. 4) and S2 (related to Fig. 7).
Comment 5. The thermal stability dataset is not completed. Figure 5 presented thermal stability data for only 5 mutants (D296R, G297A, ΔG297, D303R, T304A). To definitively rule out stability
artifacts as a confounding factor in binding assays, data for the other mutants should be
provided, even if no significant changes were observed for them.
Response 5. We agree with the reviewer that thermal stability of the protein is an important consideration in every mutagenesis study. The fact that out of 52 measurements of mutant binding (Fig. 4), 25 showed an increase, 17 no change, and only 10 a decrease suggests that introduced mutations did not significantly affect the thermal stability of arrestin-1. Nonetheless, we directly tested the stability in assay conditions (for up to 30 minutes, even though the binding assay is only 5 minutes) focusing on mutations that, based on our previous experience, change the thermal stability of arrestin-1. We did not find any dramatic effects of these mutations (Fig. 5). However, while these data indicate that potential reduction of stability did not affect our results, they do not necessarily mean that these mutations do not affect long-term stability of arrestin-1.
Reviewer 2 Report
Comments and Suggestions for Authors
The authors present the effect of substituing 17 residues on bovine arrestin-1 to evaluate its affinity for P-Rh and Rh. The paper is clear and well written. The conclusions are supported by the results.
I do not have any particular observation about the results, but it would be ideal to include in the discussion the molecular effect of the mutations in order to explain the increase or decrease on the P-Rh and Rh affinity. The authors mention it superficially, but they do not describe the conformational changes or the changes in the intramolecular interactions of the protein.
A minor observation is that the first reference to activated phosphorylated rhodopsin is described as (R-Rh*), but it should be (P-Rh*).
Author Response
Comment 1. The authors present the effect of substituing 17 residues on bovine arrestin-1 to evaluate its affinity for P-Rh and Rh. The paper is clear and well written. The conclusions are
supported by the results.
Response 1. Thanks! We highly value professional praise of the reviewer.
Comment 2. I do not have any particular observation about the results, but it would be ideal to include inthe discussion the molecular effect of the mutations in order to explain the increase or decrease on the P-Rh and Rh affinity. The authors mention it superficially, but they do not
describe the conformational changes or the changes in the intramolecular interactions of the
protein.
Response 2. Thanks! We tried to keep speculations to a minimum. However, we added some discussion of these points in the context of the conclusions regarding the role of three positively charged residues in the gate loop of arrestin-2 in binding hyper-phosphorylated C-terminal peptide of V2 vasopressin receptor (Kim et al., 2024).
Kim, K., Ashim, J., Ham, D., Yu, W., & Chung, K. Y. (2024). Roles of the gate loop in beta-arrestin-1 conformational dynamics and phosphorylated receptor interaction. Structure, 32(9), 1358-1366.
Comment 3. A minor observation is that the first reference to activated phosphorylated rhodopsin is described as (R-Rh*), but it should be (P-Rh*).
Response 3. Thank you for catching this typo! Corrected.
Round 2
Reviewer 1 Report
Comments and Suggestions for Authors
Many thanks for the author's efforts, I have no further questions.
However, I notice that those letters labelling individual panels are too big. Please modify the figures following the journal's suggestion.